# Transforming Motivation for Exercise in a Safe and Kind Environment—A Qualitative Study of Experiences among Individuals with Type 2 Diabetes

**DOI:** 10.3390/ijerph19106091

**Published:** 2022-05-17

**Authors:** Susanne Grøn Nielsen, Julie Hagstrøm Danielsen, Helle Nergaard Grønbæk, Stig Molsted, Sandra Schade Jacobsen, Tina Vilsbøll, Annemarie Reinhardt Varming

**Affiliations:** 1Steno Diabetes Center Copenhagen, Copenhagen University Hospital, 2730 Herlev, Denmark; julie.hagstroem.danielsen.01@regionh.dk (J.H.D.); helle.nergaard.groenbaek@regionh.dk (H.N.G.); ssj@nfa.dk (S.S.J.); tina.vilsboell.01@regionh.dk (T.V.); annemarie.varming@regionh.dk (A.R.V.); 2Department of Endocrinology, Nordsjællands Hospital, 3400 Hillerød, Denmark; stig.moelsted@regionh.dk; 3Department of Clinical Medicine, Faculty of Health and Medical Sciences, University of Copenhagen, 1165 Copenhagen, Denmark

**Keywords:** qualitative research, type 2 diabetes, exercise training, motivation, patient acceptance of health care

## Abstract

Exercise is a cornerstone in diabetes care; however, adherence is low and sustaining physical activity remains a challenge. Patient-centered diabetes self-management education and support are recommended; however, sparse literature exists on how to design exercise interventions that improve self-management in individuals with complications of type 2 diabetes mellitus (T2D). We aimed to gain insights into needs, barriers, and motivation based on experiences with exercise participation among individuals with T2D and complications to adjust and develop new types of tailored, supervised exercise classes in specialized care at three hospitals in Denmark. In keeping with a constructivist research paradigm, a qualitative hermeneutic approach using focus group interviews was applied to explore perspectives among different participants in terms of disease severity. Seven interviews with 30 participants (aged 49–88) representing seven different exercise classes, were conducted over three years. Reflective thematic analysis was used. Four themes were generated: People like us, Getting started with exercise, Game changers, and Moving forward. An overarching theme ‘The transformation of motivation when exercising in a safe and kind environment’ links the themes together, resembling the participants’ development of physical literacy encompassing motivation, confidence, physical competence as well as an ability to value physical activity. Supportive patient-centered exercise classes promoted a transformation of motivation grounded in the development of physical literacy among participants in specialized diabetes care. However, participants were concerned with continuing to exercise on their own after the intervention, as they experienced a lack of continuous, supervised exercise opportunities in local communities.

## 1. Introduction

Type 2 diabetes (T2D) is a chronic metabolic disease reducing quality of life and life expectancy [1,2]. T2D is strongly associated with unhealthy eating habits and a sedentary lifestyle, leading to insulin resistance, hyperglycemia as well as macro- and microvascular complications [1]. ADA/EASD [2] and WHO [3] recommend multifaceted interventions in the (self)management of T2D comprising diet, physical activity, medication, smoking, and stress management. Although the different elements should be coordinated and aligned in practice, individual elements need to be explored and developed separately for specific target groups to enable the implementation of best practices. The present study focuses on enhancing physical activity among individuals with T2D and complications. Supervised exercise training with combined aerobic and strength training improves glycemic control [4,5] and is recommended as a cornerstone in diabetes care [2]. Like populations in general, individuals with T2D are advised to adopt 30 min of daily physical activity [6], however, adherence to intervention participation is low and sustaining lifestyle changes remains a challenge [7,8].

Sparse knowledge exists on the design and contents of exercise interventions that promote participation, motivation, and adherence to lifestyle changes with physical activity. International guidelines recommend that lifestyle changes are promoted through patient-centered diabetes self-management education and support; a process aimed at helping people with diabetes to gain the knowledge and skills necessary to navigate decisions and activities in diabetes self-care [2]. Self-management implemented in a patient-centered manner that is respectful of personal preferences, needs, and values is hypothesized to improve diabetes-related outcomes [7,9].

Patients’ perspectives on physical activity participation have been investigated in qualitative studies in individuals with T2D. Aiming to understand factors influencing engagement in physical activity a scoping review found that in particular motivation and social support affected physical activity behaviors [10].

In Denmark, specialized diabetes care for individuals with complications is located at hospitals where supervised exercise is not offered systematically, and no theoretical framework exists for the design of exercise interventions that are accepted by individuals with complications of T2D. Therefore we aimed to gain insights into needs, barriers, and motivation based on experiences with exercise participation among individuals with complications to T2D to adjust and develop new types of tailored and supervised exercise classes in specialized care at three hospitals in Denmark.

## 2. Materials and Methods

### 2.1. Study Design and Ethics

This qualitative study was embedded in a three-year supplementary treatment initiative with physical exercise in specialized diabetes care at Steno Diabetes Center Copenhagen, Herlev-Gentofte Hospital and North Zealand Hospital in Denmark. The treatment initiative aimed to develop patient-centered exercise interventions that targeted individuals with complications of T2D [11]. In keeping with a constructivist research paradigm, a qualitative hermeneutic approach using semi-structured focus group interviews was applied [12,13]. The focus group interviews were conducted at different time points to inform the development of different types of exercise classes, and during this process, it became apparent that certain experiences and perspectives were comparable across the dataset, which formed the analysis presented in this qualitative study. In Denmark, no ethical approval is required in qualitative studies. The trial was performed in accordance with the Helsinki Declaration and registered at the Danish Data Protection Agency (VD-2018-480; I-Suite: 6739).

### 2.2. Inclusion Criteria

Participants were consecutively referred to the exercise intervention by health care professionals at regular control visits. Inclusion criteria were age ≥18 years, T2D with glycemic dysregulation, and/or increased risk of complications based on values of glycated hemoglobin A1c (HbA1c), blood pressure, and albuminuria according to level 2–3 in the Danish risk stratification model [14].

### 2.3. Intervention

After referral, individual consultations were offered by a physiotherapist or an exercise physiologist to establish eligibility for participation and select the most suitable exercise class. Inspired by the concept of motivational interviewing; motivation and barriers to physical activity were explored [15]. Physiological and patient reported outcomes were assessed and results are described elsewhere [11]. Twenty-four supervised group-based exercise sessions with combined aerobic and strength training were offered. A variety of classes were available encompassing different functional levels, geographical locations, and times of day. In classes tailored to individuals with low functional levels or vulnerable individuals, i.e., women with ethnic minority backgrounds or individuals with neuropathy, class sizes were reduced from twelve to eight participants per class to enable individualized support. Some classes were structured with continuous uptake which is a cost-effective solution allowing an efficient turnover increasing the occupancy rate and availability for new participants. Other classes had a fixed start and finish to enable the delivery of structured exercise plans with a graduate introduction to physiological load and to encourage the building of social relations. Individual exercise programs and support were offered to individuals unable to participate in a supervised class due to work, transportation, or other reasons. Experienced exercise physiologists or physiotherapists supervised all exercise sessions applying patient-centered treatment and support in handling any challenges participants may experience. Participation was free of charge and assisted transportation between home and hospital was offered.

### 2.4. Focus Group Interviews—Data Collection

Seven different classes were evaluated with focus group interviews, in an ongoing process from February 2018 to September 2019 (Table 1). Characteristics of participants in the exercise intervention are described elsewhere [11].

The first two explorative interviews were conducted in February 2018 with participants in two already existing exercise classes at Herlev-Gentofte and North Zealand Hospitals. These classes had a “one-size fits all” design with a continuous uptake and researchers (SGN and JHD) had no prior knowledge of or relation to these participants. In July 2018, three evaluative interviews were conducted after completion of the first round of the newly designed classes: The Soft Start class, The Morning class, and The Outdoor class. SGN and JHD were familiar with the participants; however, the interviewer was not in charge of the particular exercise class being interviewed. Further two interviews were conducted in September 2019 evaluating the Women Only class and the Neuropathy class. A student-assistant not familiar with the participants conducted these interviews. Class sizes were small with six to twelve participants. In classes with a fixed start and finish the interviews were conducted at the end of the 12-week course. In classes with a continuous uptake, the participants had participated in between two to 24 sessions at the time of the interviews. A semi-structured interview guide was constructed by SGN and JHD in 2018, with minor adjustments prior to interviews in 2019. Questions were open-ended encouraging reflections and dialogue between informants (Table 2). Informants were asked to keep confidentiality regarding the information revealed during the interview. Interviews were audio-recorded using a digital voice recorder (Olympus) and transcribed verbatim by JHD or SGN, whereafter the recording was erased. Transcriptions in Microsoft Word were stored in a locked digital folder in the Capital Region of Copenhagen, Denmark assessable by the research group only. In the dissemination of the results, personally identifiable data were omitted.

All participants in the classes evaluated were invited to participate in focus group interviews with oral and written information. Approximately one week’s notice was given to consider participation and, after written consent, interviews were held in an undisturbed room next to the gym at the hospital. For convenience, the interviews were conducted 15 min after the exercise class allowing participants time to shower and revitalize. The purpose of the interview with information on the confidentiality of data was explained at the beginning of the interviews, followed by a presentation of the interviewer and participants. The interview guide was followed informally inviting any reflections and discussions among participants. In the end, all were invited to comment on the process of the interview. There were no withdrawals during the interviews nor afterward.

### 2.5. Data Analysis

All interviews were analyzed as one material using NVivo, ©QSR NVivo12. In the theoretical framework of interpretivism, the reflexive approach of thematic analysis was applied providing a six-stage systematic, rigorous and comprehensive process (Table 3) which is a flexible method for qualitative analysis, enabling an inductive approach with freedom in applying philosophical and theoretical frameworks [16,17,18]. The initial coding and formation of themes were conducted by the first author, however, throughout the analysis, the findings were reviewed and refined with all co-authors and adjusted accordingly.

## 3. Results

Fifty-nine individuals were assigned the seven classes of which thirty accepted the invitations and participated in an interview. Reasons for not participating were depression, lack of time, i.e., going to work or issues with transportation. The number of the participants in the interviews was also affected by the rate of attendance in the exercise class on the day of the interview. Throughout the intervention period the attendance rate was 71% [11]. Moreover, the interview with the lowest rate of participants (three out of 10) was a class with a continuous uptake, which may have affected the motivation for participating in the interview. Half of the participants were women and the mean age was 66 years (range 49–88). The morning class had the youngest participants, with two still working, whereas all others were retired or early retirees (Table 1).

The initial analysis generated six themes with 17 subthemes (see Appendix A). Reviewing and refining themes gave rise to a main overarching theme ‘The transformation of motivation in a safe and kind exercise environment’. This overarching theme illustrates a journey through different experiences of motivation from doubtfully accepting referral to completion with satisfaction and incipient confidence in the ability to self-manage physical activity. In keeping with the overarching theme, four themes and eight subthemes were generated to embrace participants’ reflections on and experiences with initiating and adopting physical activity in daily life (Table 4).

### 3.1. People like Us

The participants described themselves as individuals with specific characteristics as well as groups of people with similar life circumstances when suggesting reasons for developing diabetes and when describing behavioral, physical, and mental aspects associated with living with T2D. Even though participants, thus, having had concerns in relation to starting to exercise, they accepted the invitation. Some had previous positive experiences with exercising and others were convinced, but most wanted to become healthier or reduce weight.

#### 3.1.1. Living with Type 2 Diabetes

The participants described a broad spectrum of bodily challenges such as persistent knee pain or recurring foot ulcers, as well as mental issues such as depression affecting their daily living and diabetes care. Musculoskeletal problems were described by many participants as barriers to physical activity. There was a common understanding or recognition that many individuals with T2D suffer from joint and muscle problems as one participant stated:

“You know, we all have something to deal with, you get a frozen shoulder and then you get a bad knee because the mucosa is bad” (woman, age 66 years)

Moreover, they shared reflections on how a sedentary lifestyle, had contributed to the development of diabetes:

“There may be a reason why we are the ones who got diabetes, right? It’s probably because we weren’t that damn good at moving around and having a high self-discipline. Folks like us… I should have exercised more” (woman, age 61 years).

#### 3.1.2. Reasons for Starting to Exercise

Meanwhile, joint problems, overweight, poor fitness, or hyperglycemia were reasons for engaging in the exercise program. The motivation for accepting referral and initiating exercise training was driven by both external factors and internal factors. Many were encouraged by their nurse, physician, or family members. As one woman explained:

“I have had diabetes for about 10–15 years, and I am here because I have become too big and round and I could use some exercise. My legs have become stiff. So, the doctor, the endocrinologist, thought that it would be a good idea if I signed up, and I said, that I thought so too” (woman, age 69 years)

Internal motivators were, for example, a longing to take up sports that they had played at a young age or as one man explained, a desire to change the desperate feeling of losing physical function:

“I came here because I don’t walk very well. And I have hardly any sensibility in my feet. Also, I have trouble with my feet. I cannot use ordinary sandals, so I trip, and fall and hurt myself” (man, age 66 years).

### 3.2. Getting Started with Exercise

Worries about being too unfit to take part in the exercises and fear of pain or joint problems were expressed by several participants. The impact of fear of pain and other harm from exercise activities was substantial and made participants emphasize a need for professional support when starting and implementing exercise. Furthermore, participants highlighted the importance of a safe and kind environment in terms of not feeling ashamed or being different from peers.

#### 3.2.1. Feeling Safe and Daring—The Role of the Therapists

Being met by dedicated health professionals offering individualized attention and support was very important for daring to join the intervention and exercise activities. Participants emphasized that the professionalism of the multidisciplinary team with knowledge of complications of diabetes, and the structured exercise program with tailored exercises increased their confidence, as described by this participant:

“They [the physiotherapists/exercise physiologists] were really good at… if they give you an exercise… now I have trouble with my feet and crawling on the floor is not a good idea, but immediately they have an alternative. They don’t let you off, and they shouldn’t. They always suggest some other way. There is no chance of using excuses for taking a rest (laughs)” (man, age 62 years).

Participants highlighted that individualized attention and kind support endorsed a feeling of safety both physically and mentally:

“So feeling safe is important? (interviewer, red.).” Definitely, they [the physiotherapists/exercise physiologists] are very attentive. If we stop because of pain or exhaustion, then … you are very kind to us… what is going on now? And this is very, very important. You feel safe… you know? If I went to the fitness center, no one would keep an eye on me” (woman, age 62 years).

#### 3.2.2. Feeling Safe and Daring—The Role of Peer Support

By meeting peers, participants shared experiences concerning their diabetes management and complications. This was valued both by those newly diagnosed and by those having had T2D for a long time, as for example:

“It’s good to hear others’ experiences and such, I haven’t been in this for very long, this diabetes, so it’s nice to talk with others with diabetes. How they are doing, if they use insulin, kind of to monitor how far they have come…” (man, age 62 years).

Further, belonging to a group and being accepted despite their disabilities was important. The friendly atmosphere motivated participants to come to class, for example, one participant said:

“But I think that the way that we played games in the beginning, we got to know each other and learned each other’s names… suddenly we were a team. If you had been in the municipality where they come and go every week, then it would just have been “hello and goodbye”. You know, they gave me the feeling that we were here as a team” (man, age 74 years).

Participants from a class with continuous uptake discussed the disadvantages of the constant turnover, described as “revolving doors” where close relations did not develop.

“Those relations—they don’t become so close when the ‘revolving doors’ are turning all the time. So, if we were to have closer relations after training, if that was a goal with the training sessions, then it should probably be organized in a different way” (woman, age 73 years).

### 3.3. Game Changer

Experiences of becoming fit and recognizing positive changes in their health status led participants to changes in their motivation for exercise. Furthermore, the experiences of learning how to exercise without adverse events endorsed confidence and promoted the change in motivation which became a turning point—a game changer.

#### 3.3.1. Bodily Experiences

Feeling stronger and seeing changes in body composition or fitness tests were emphasized as motivational factors, for example, as expressed in the following:

“You actually feel it in your body, that you are better, right? I have been very frustrated over gaining weight, but I know now that it is muscle. So, it is good, I have lost weight, but primarily I have gained muscle mass” (woman, age 61 years).

Participants described how they learned to increase intensity during aerobic exercises; thus, gaining both skill and experiences of well-being, as stated by this participant:

“I have learned that it is good to sweat—and it feels good as well” (man, age 62 years).

Plasma glucose was measured before and after training sessions in one class and one participant described how the blood glucose measurements were a relief in his diabetes self-management, as he did not have to struggle with it at home. Another participant described how it motivated him:

“I think it’s fantastic, when you come it’s over 7, but then it is 5.9 [mmol/L] when you go home” (man, age 68 years).

#### 3.3.2. Increased Self-Confidence

Participants expressed relief and joy with having been able to exercise without pain or strain. Improved physical function changed their attitude towards physical activity and their confidence in being able to exercise, which motivated them to do more.

“I have learned that I dare now, I have had a long period of time not daring... because I was afraid of what could happen to my back and my neck. I have learned, for sure, that I can do more than I thought. I can challenge myself more than I thought. That makes me happy.” (woman, age 61 years).

Some participants described with satisfaction how they could manage activities of daily living again.

“So, I am pleased with those exercises lying on a mat. It’s my first time on the floor. Down there and having something to pull myself up by. It has given me more self-confidence with falling and such—with what I can do” (man, age 61 years).

Great satisfaction was expressed with their achievements, having participated, made it through, and then seeing the results was very motivating:

“All of a sudden, you can walk an extra 200 m in the same time, that’s really nice” (woman, age 73 years).

### 3.4. Moving Forward

Self-determination and feelings of being empowered were expressed, along with concerns about their ability to continue with physical activity independently after the intervention. Plans were made for how to stick with the lifestyle changes.

#### 3.4.1. Renewed Motivation

Participants reflected on feelings of motivation, that they had not felt before. Having gained skills and experienced health benefits, they felt motivated with confidence that exercise participation was possible. This strengthened their determination regarding physical activity:

“I hate it, but I come here voluntarily. I cannot accept having a chronic disease. It’s my health and I have to do something about it” (woman, age 62 years).

“Definitely, we have learned which exercises to do at home. You can do some self-care at home. I haven’t been motivated to do it before, but I am now” (man, age 88 years).

Participants explained how they exercised at home, having made plans, and taken initial steps for maintaining physical activity on their own, for example:

“I do a lot of sit to stand exercises. I have placed a chair out in the kitchen, so when I put the kettle on for coffee or tea, or whatever I am making, while it’s boiling, I sit and rise from the chair. It’s those little things that I do at home. Not as much as I should, but I do something.” (woman, age 69 years).

#### 3.4.2. Having Concerns

Difficulties with changing habits and sticking to lifestyle changes were also discussed. Participants who were facing the upcoming challenges of exercising on their own argued, that 12-weeks of supervised training was too little time for establishing sustainable behavior changes. Further, they elaborated on practical challenges such as the lack of access to facilities with proper equipment and professional guidance, as well as unaffordable expenses and the need for training partners:

“I think it should last a bit longer. It has taken me 10 years to get started with doing sport. It took a diabetes diagnosis and a bit of nudging as well. And then it only lasts three months and that time is gone like this (snaps fingers). And there you are again all alone, and I am scared as hell that if I don’t find some allies then it won’t happen” (woman, age 61 years).

## 4. Discussion

### 4.1. Transformation of Motivation

An overall finding in the present study was that motivation seems to transform from an initial state of uncertainty into a stronger state of determination, a process that summarizes and links the results of the data analysis. In the theme ‘People like us’, participants described a vicious circle of low fitness, overweight and musculoskeletal problems with worries about adverse events and thus a hesitation to apply the recommended physical activity, but for different reasons accepted to participate in exercise classes. Due to feeling safe with the professionals and peers, participants dared to exercise which allowed for positive bodily experiences and increased self-confidence with exercise performance. This facilitated participants’ motivation for continued exercising after the supervised exercise intervention, however, at the same time they were expressing concerns regarding being on their own in the future. This process resembles the flow in the motivation continuum from amotivation through extrinsic motivation to intrinsic motivation described by Ryan and Deci [19]. Similar results were found in the study by Sebire et al. in people with newly diagnosed T2D [20], indicating that such processes are independent of the stage of disease or severity, and possibly more dependent on individual circumstances and stages of change, and may be facilitated by appropriate resources and settings.

### 4.2. People like Us—Starting to Exercise

Fear of harm is a well-known barrier to exercise participation among people with complications of diabetes or co-morbidities [21]. In the present intervention, 7% of the participants experienced an adverse event that emphasizes the actual risk and explains participants’ need for supervision and support and reinforces the importance of endorsing a feeling of safety and daring to join in on exercises. Further, the supportive atmosphere and kind attention from the health professionals seemed to facilitate the management of mental challenges such as pain, fear, or shame. These findings are consistent with the results presented by Olesen et al., who found that diabetes self-management was best achieved when taught in a patient-centered manner respectful of personal preferences and needs [9], which might be even more important in this target group of individuals with complications to T2D [7,14]. Interestingly, the terms ‘individualized treatment’ and ‘patient-centered care’ are described by Cochrane et al. as ‘compassion’ [22], which in a health care context is characterized as a professional relationship with a genuine desire to understand a person’s needs leading to qualified clinical decisions. It is argued that organizations that manage to offer health care in a compassionate culture have the potential to achieve positive patient experiences with higher treatment safety and quality [22]. Furthermore, in the present study, peers were experienced as buddies with whom participants could feel alike and share stories, experiences, learn about diabetes management, and establish training communities, which were highly appreciated and highlighted as an important element. This finding resonates with several other studies [10,23,24,25,26], which found peer support to be crucial in behavior change processes. Peer support has been described to contribute to creating comfortable non-stigmatizing exercise environments and studies have shown that group-based diabetes interventions are beneficial for sharing experiences and social support [19,27]. Social relations are suggested to serve as basic psychological needs important to promote motivation and self-efficacy [19,28].

### 4.3. A Game Changer—Developing Competences, Confidence, and Renewed Motivation

The experiences of satisfaction, enjoyment, skills, or accomplishment described by participants in the present interviews, are in alignment with central components of intrinsic motivation described in The Self-Determination Theory and believed to be drivers of a more stable and stronger motivation than advice or pressure [19]. A Finnish study specifically focusing on determinants for pursuing and maintaining physical activity, found strong associations between the experience of receiving autonomous support to establishing autonomous motivation towards physical activity [29]. Such associations may explain how the transformation seen in the present study occurred, with a change from an externally motivated participation (doctor advice or an obligation) towards a genuine autonomous motivation as competencies develop and self-confidence increases. In line with these findings, a scoping review [10] identified self-confidence and self-efficacy to be important factors for physical activity in adults with T2D. Moreover, a study investigating exercise barriers and the relationship to self-efficacy for exercise among people with heart disease and/or diabetes found self-efficacy to be associated to motivation, thus highlighting the importance of enhancing participants’ self-efficacy [30].

### 4.4. Linkage to the Framework of Physical Literacy

The above-mentioned findings in terms of competencies, confidence, and motivation are core elements in the concept of physical literacy, which presents basic elements necessary for adopting, accepting, and sustaining physical activity [31]. Physical literacy is defined as: ‘the motivation, confidence, physical competence, knowledge and understanding to value and take responsibility for maintaining purposeful physical pursuits/activities throughout the life course’. It is argued that the development of physical literacy is a prerequisite for adopting physical activity [32]. Sparse knowledge exists on how to design exercise interventions promoting Physical literacy in adults with T2D, however, as indicated in the present study, offering tailored exercise interventions, where participants dare to experiment, e.g.,: ‘It’s my first time on the floor’, may promote physical literacy. Improvements in physical literacy were measured in a study targeting physically inactive adults [33]. However, a validated tool for measurement of physical literacy is not available yet and Holler et al. suggest further work in this regard [33]. Future studies, both qualitative and quantitative, are needed to further explore how to promote physical literacy among individuals with T2D.

### 4.5. Moving Forward—Future Perspectives

As illustrated by the subtheme ‘Having concerns’ participants did not feel comfortable and ready to be on their own in continuing exercising after the intervention period. Several other studies have found similar results and advocate that programs should enable linking physical activity opportunities to participants’ everyday life practices [26,34]. Generally, there seems to be a gap in sectorial transitions from being part of supervised exercise interventions and having access to facilities for independent exercise. This study revealed that individuals with T2D and complications can be engaged in physical activity and, therefore, as all individuals with multimorbidity, will benefit from collaboration between hospitals, municipalities, and private fitness centers and sports clubs in offering safe access to basic physical exercise training, which is supported by health professionals according to individual needs.

### 4.6. Strengths and Limitations

A strength of this study was the number of interviews conducted at different timepoints covering seven different classes with 30 individuals with complications to T2D—a rarely investigated target group. Moreover, the study provided comprehensive insights into the experiences with exercise participation. Additionally, a strength was that participants were recruited from three hospitals covering a large geographical area with both rural and urban citizens, as well as varied socio-economic statuses representing the population broadly. The process of coding and the development of themes address and reflect participants’ experiences across different exercise classes with a limited number of participants per class. However, this also introduces a limitation regarding the use of the data with respect to each specific exercise class. Furthermore, the results are not generalizable nor directly transferable to other settings and populations. Moreover, the interviews were conducted originally with an evaluative purpose, so interview guides were not explicitly constructed to explore motivation. A further limitation was that approximately half of the participants declined participation in the interviews, thus their experiences are not represented.

## 5. Conclusions

This qualitative study revealed insights into important factors when designing interventions to increase physical activity in individuals with complications to T2D as described by the four themes: People like us, Getting started with exercise, Game changers, and Moving forward. The intervention enabled to engage individuals with complications of T2D and a sedentary lifestyle in physical activity by developing tailored, supervised exercise classes. The overarching theme, ‘The transformation of motivation when exercising in a safe and kind environment’ is linking the themes and demonstrates a change process from being externally motivated with feelings of insecurity to the development of self-confidence and internalized motivation, resembling the development of physical literacy. However, the study also revealed a major drawback as participants were concerned with regard to continuing exercising on their own after the intervention, emphasizing the need for professional and social support. The findings are of importance from a public health perspective since there exists a gap between the recommendation of a physically active lifestyle among individuals with T2D and a lack of continuous, supervised exercise opportunities in participants’ local communities integrated with their everyday life.

## Figures and Tables

**Table 1 ijerph-19-06091-t001:** Interview and participant characteristics.

Interview	Class Size	Duration, min (Pages)	*n*	Female, *n*	Age (Range)
Original Herlev-Gentofte	10	51 (18)	3	2	65 (61–73)
Original North Zealand	6	42 (21)	4	2	72 (68–80)
Soft start class	8	18 (8)	5	0	70 (63–80)
Morning class	6	26 (12)	3	2	61 (55–67)
Outdoor class	6	25 (12)	5	4	62 (49–67)
Women only class	7	25 (18)	3	3	62 (52–68)
Neuropathy class	8	35 (14)	7	2	72 (62–88)

Abbreviations: min, minutes; *n*, number.

**Table 2 ijerph-19-06091-t002:** Interview guide.

1. Please introduce yourselves—why are you attending this diabetes exercise class?
2. What are your experiences—has anything surprised you positively or negatively?
3. How have your participation changed anything for you? What have you learned?
4. What helped you to succeed with coming here and doing exercises?
5. How has it been for you to work hard or sweat or feel exhausted?
6. How has the blood sugar measurements made any difference for you?
7. How do you plan to continue physical activity in the future?
8. Do you have any suggestions for changes in future exercise classes?
9. Is there anything else you would like to comment on?

**Table 3 ijerph-19-06091-t003:** The thematic analysis.

Phases	Description of the Process
Data familiarization	Transcribe data, read and re-read, note initial reflections.
2. Generate initial codes	Code interesting elements across the entire dataset.
3. Search for themes	Organize codes in potential themes, gather all data relevant to each theme.
4. Review of themes	Discuss themes in relation to the dataset and theory. Generate a thematic map.
5. Final themes	Ongoing analysis of findings across the dataset. Translate quotes. Define themes.
6. Report	Present results, discussion, and conclusion.

**Table 4 ijerph-19-06091-t004:** Final themes and subthemes.

	Transforming Motivation in a Safe Environment			
Themes	People like us	Getting started with exercise	Game changer	Moving forward
Subthemes	Living with type 2 diabetes	Feeling safe and daring—role of the therapists	Bodily experiences	Renewed motivation
	Reasons for starting to exercise	Feeling safe and daring—role of peer support	Increased self- confidence	Having concerns

## Data Availability

Not applicable.

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
