# Peer review of "Transforming Motivation for Exercise in a Safe and Kind Environment—A Qualitative Study of Experiences among Individuals with Type 2 Diabetes"

_ijerph, 2022, doi:10.3390/ijerph19106091_

Round 1

Reviewer 1 Report

The authors developed a qualitative study aimed to obtain information about "needs, "barriers, motivation, and experiences perceived and expressed by diabetic patients enrolled in an exercise program in three Danish Hospitals.

First of all, I'd like to say that I consider this paper a very good paper, addressing a very important subject and bringing new perspectives. The presented study is also veyr good. Nevertheless, some comments follow:

Strengths

Adhesion to exercise programs is a very relevant topic in diabetic patients' care; it's usually low, but considered important, In this study several reasons have been identified pointing to new approaches. Until now no sustainable solutions have been found., The new ideas need to be further explored to become operational. 
The use of qualitative methodology allows the expression of  subjectivity and exploration of details not well understood, as in this study
Regarding demography, the age span is large (49-88) eventually with younger ages not represented
Context is described and allows a good perspective of the people and conditions involved
Themes and subthemes identification are clear and operational for future studies 
Although not being developed for the study patient-centered, ongoing exercise programs used are a clear strength with an interesting diverse approach

Weaknesses
As the exercise programs have been developed for three years it would be interesting to consider changes in the same people's views and perspectives a long time
The use of 7 different exercise programs and the heterogeneity of the groups (people at the beginning of the program mixed with those participating in up to 24 sessions) can introduce some problems in interpretation
only one author coded the interviews, 
The role of the health care professionals involved and the quality of communication they establish with patients was not addressed directly in the questionnaires although from several patient statements seems to be important. (line 199-200)

line 99 "-" not needed in "sup-port"

Reviewer 2 Report

First of all, I want to congratulate the authors because it is a subject that has been little studied in public health and is very necessary at this time. Many ways have been tried without results to improve the quality of life of people with chronic conditions. However, regarding the writing of the article itself, it presents severe omissions regarding methodology and design that are essential in qualitative research. In an improved version of this, it is hoped that all the theoretical support of this research can be evidenced and give coherence to the complete work developed.

Title: Although the title invites reading, it does not represent the research carried out or its contribution to public health. It is suggested to look for a title that shows more of the study than calls for reflection that is sought to be achieved.

Abstract: The research paradigm is not specified, nor is the focus of the study on the qualitative design, which is vital for the research's coherence.

MeSH: experiences are not a MESH descriptor. Check the relevance of all words.

Introduction: The delimitation of the object of study must be improved since the objective of the research speaks of needs, barriers, motivation, and experiences regarding participation in the exercise; but only the need to perform physical activity is raised without trying to justify the type of research to be developed or the elements that it studies. It should also be improved that almost 40% of the references in this section are more than ten years old. In addition, the main actors in this issue at the global level, such as WHO, are not included.

Methodology: The research paradigm is not specified, nor is the focus of the study on the qualitative design, which is vital for the research's coherence. That is, knowing if the data collection method is appropriate and the data analysis is consistent with the phenomenon to be investigated. There is no clear description of the subjects to be investigated regarding the population. The guiding questions are not detail how their construction and validation took place. There is no explanation for the small number of participants in the focus groups or how they are carried out. It also remains to specify whether the researchers had a previous relationship with the participants and how the contact with them was made. In the data collection, the description of the place where the interview was conducted, the recording method, and whether there were refusals to participate in the research or withdrawals in its development. It does not specify how the confidentiality of sensitive data was maintained and which theoretical reference was used as the author for the thematic analysis.

Results: some narratives should be revised based on the definition given both the subtopic and the topics themselves.

Discussion: the subtitles placed in the sections are not clear. It is suggested to use the same ones as in the results and follow coherence. It should be increased since a small discussion is not consistent with the number of focus groups and the results presented. Likewise, it is necessary to integrate more primary articles in the discussions at an international level and from the last five years.

Conclusions: it only provides a summary of the findings. Still, it does not mention whether the objective was met in the face of the chosen methodology or what is the contribution of this research to public health.

References: These must be updated and include the main actors worldwide, such as WHO, because although approximately 64% are from the last five years, almost 30% are more than ten years old.

Round 2

Reviewer 2 Report

First of all, once again, congratulations to the authors for the theme and the approach taken. Although the article's writing has been improved, there are still some topics pending in methodology that need to be improved for publication.

Title: improved.

Abstract:

No research paradigm is specified, but the rest has been improved.

MeSH: Improved.

Introduction: The delimitation of the object of study and the references have been improved.

Methodology: The research paradigm is not specified. Nor is there an explanation for the small number of participants in the focus groups. It also remains to establish whether there were refusals to participate in the research or withdrawals in its development. It does not specify how the confidentiality of sensitive data was maintained and which theoretical reference was used as the author for the thematic analysis.

Results: It has been improved.

Discussion: It has been improved.

Conclusions: It has been improved.

References: It has been improved.
